# The Future of Accounting: How Will Digital Transformation Impact the Sector?

Maria José Angélico Gonçalves [1,*] , Amélia Cristina Ferreira da Silva [1] and Carina Gonçalves Ferreira [2]

1 CEOS.PP, Polytechnic of Porto, Instituto Superior de Contabilidade e Administracao do Porto (ISCAP), 4200-465 Porto, Portugal; acfs@iscap.ipp.pt
2 Polytechnic of Porto, Instituto Superior de Contabilidade e Administracao do Porto (ISCAP), 4200-465 Porto, Portugal; carina-goncalves@live.com.pt
* Correspondence: mjose@iscap.ipp.pt; Tel.: +351-229050000

**Abstract:** The growing dissemination of digital technologies has had an incomparable impact on many dimensions of today's civilisation. Digital transformation (DT) redefined the industrial structures and reinvented business models. Hence, in the face of Industry 4.0, financial and accounting services face new threats, challenges, and opportunities. How do the business players in the accounting sector perceive this phenomenon? This paper aims to answer this question by following a qualitative and exploratory approach, applied to three case studies, using semi-structured interviews. The study shows that although digital transformation in Portuguese small and medium-sized accounting service enterprises is just starting, Industry 4.0 technologies, optical character recognition (OCR), artificial intelligence (AI), robotics and enterprise resource planning (ERP) in the cloud were the technologies singled out by respondents. Resistance to change, organisational culture and price seem to be the main barriers to DT in accounting. This paper contributes to a better understanding of the role of accounting and accountants in organisations and society in the context of the digital era. Moreover, it provides practical insights into the potential relationship between technological (specifically digital) development and labour market dynamics for accounting professionals.

**Keywords:** accounting; digital transformation; Industry 4.0; digital skills; information technology

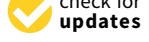

## 1. Introduction

Digital transformation is characterised as the merging of physical and digital processes into decentralised systems, representing a major change in the social and organisational environment. It affects all aspects of people's lives, as organisations and society changed their structures, and economy and businesses reinvented their value creation process. Accounting is no exception. According to Jasim and Raewf (p. 55, [1]) "the use of information technologies to simplify accounting processes and reduce the effort of the accountant started more than 140 years ago".

The impact of digital transformation in accounting is still very unclear. The access to distributed ledgers (blockchain) and Big Data, supported by cloud-based analytical tools and artificial intelligence, will automate decision making on a large scale [2–5]. However, automation also increases the risk of jeopardising the quality of information [6]. Thus, although the abundance of data suggests that the decision-making process is more rational, at the same time, there is an increase in complexity [7].

From the perspective of accounting professionals, DT can be seen as threatening, since IT allows for the automation of activities and working processes that were assumed by them. However, it can also be seen as an opportunity, since it releases accountants from those boring tasks that can be performed by a machine, giving them time to focus on more value creation activities.

This study aimed to analyse the impact of digital transformation in the field of accounting from the perspective of the leader in the sector, i.e., the business of accounting. More

specifically, it aims to: (1) identify the level of implementation of Industry 4.0 in Portuguese companies providing accounting services; (2) identify the main benefits, challenges, and barriers to the implementation of Industry 4.0 in the accounting sector; (3) explore the impact of Industry 4.0 on accountants' daily life and the future of the accounting profession; (4) identify the cybersecurity risks of accounting information due to the use of the emerging IT; and (5) identify the digital skills of accounting professionals operating in the digital era.

For this purpose, an exploratory study of a qualitative nature was carried out, which consisted of the analysis of three case studies, using document sources and semi-structured interviews. The interview scripts were designed for two distinct profiles: middle businesses of the value chain, i.e., back-office businesses, where IT software and consulting are the main products; and final businesses of the value chain, i.e., front-office businesses, where accounting services and tax consulting are the main products. In this study, in contrast to previous research, we present the point of view of professional accountants and IT managers regarding DT in Portuguese small and medium-sized enterprises (SMEs).

After this introduction, Section 2 presents the literature review. In Section 3, the methodologic approach, the procedures for data collection, and data analyses are described. Section 4 presents and discusses the results of the study. Finally, the main conclusion and the limitations of this study are described in the final section, in addition to future research opportunities on the topic.

## 2. Literature Review

Yoon [8] assesses the effects of IT adoption in accounting practices and the accounting profession. The author states that the accountant of the future will be the one who can embrace the future technological changes and be part of the system.

Yuksel [9] defines Industry 4.0 and its innovations in the field of accounting. He mentioned that automation of accounting information systems using Industry 4.0 technologies will contribute to real-time access to information and security. He also mentioned that processes will be executed faster and with greater transparency.

According to Berikol and Killi (p. 1, [10]) "many enterprises have begun to use modern cost and management accounting tools through institutional integrated information systems" associated with the digital transformation process.

Digital transformation is essential for a business to keep up with the market and position itself competitively. The adoption of IT allows organisations to make more efficient decisions, enabling a more agile response to any opportunity or challenge [11]. Technology stimulates change on many levels in the organisation. Indeed, as mentioned by Matt et al. (p. 339, [12]) "the exploitation and integration of digital technologies often affect large parts of companies and even go beyond their borders, by impacting products, business processes, sales channels, and supply chains".

Companies have evolved in the management process, the industry has modernised production processes, and people have evolved in the way they relate, buy, negotiate, and interact with technology. Accounting is a structural component of an organisational information system from an integrated perspective. The current information systems are the result of years of technological and procedural evolution. Guerrero and Sierra [13] identified six levels of evolution of information systems:

Level 1—Transactional—Summary reports, ERPs
Level 2—Tactical—Consolidated Reports, Excel, Access
Level 3—Analytical—Analytical Tools, KPIs, Datamart, User access
Level 4—Strategic—Predictive modelling, Dashboards, Data governance, Wide user access
Level 5—Advanced—Real-time, Advanced analysis, Scorecards, Automation, Forecast, Decision X, Events, Corporate DW, Corporate KPIs, Business, Government
Level 6—Innovative—Globalised corporate process, Analytical services, Enterprise data warehouse, Collaborative asset.

Information technologies, currently referred to as Industry 4.0, offer enormous potential to the field of accounting. Many of the tasks being performed by accountants will be

automated. Accountants are being replaced by robots in their routine tasks, creating more space for other accounting activities, such as data analysis. Since the role of accountants will continue to be decisive for organisations, despite the expected changes in their role at the workplace, Hoffman [14] suggests that accountants enrich their creativity and sense of improvisation to replace themselves with the value creation process of the organisation.

Information technology innovation contributed to the development of corporate accounting systems, improved business performance, and helped the emergence of cloud accounting. The simplification of accounting procedures while enhancing their efficiency and effectiveness due to the use of information technologies has resulted in greater opportunities for companies to expand their commercial deals and enhance public confidence in them. A previous study (p. 159, [15]) listed the main benefits of IT as being the following: "reducing the risk of error (especially human error), low risk of fraud, system automation, big data analysis, huge cost savings (by increasing the efficiency and decreasing in errors), increased reliability in financial reports and reduced workflow". However, to obtain these benefits from digital transformation, organisations must guarantee the interoperability and integration of IT solutions [16].

Despite the benefits, risks have also been identified in the literature, namely, those related to cybersecurity [17]. Yau-Yeung et al. [18] distinguish between: (a) general risks associated with cloud accounting, which correspond to those pointed out in the larger cloud computing literature, such as hardware and software compatibility, Internet/server stability, data security, and data loss; and (b) specific risks to cloud accounting systems and services: (i) financial statement reliability, (ii) legal compliance, (iii) location of data, and (iv) ownership of data. Indeed, since a cybersecurity breach could shut down an entire critical infrastructure industry and threaten organisational survival, this topic has gained great relevance in the accounting domain [19,20].

Moreover, regarding digital transformation in the context of the organisation, it is important to be aware that it is more than a simple cost–benefit analysis process, "it involves more or less profound changes in the business model of the company, which may occur on processes, resources, operational methods or culture" [21]. Often, the main obstacles to the digital transformation of organisations are intangible. As stated by Schwertner (p. 388, [22]) "The ability to digitally reimagine the business is determined in large part by a clear digital strategy supported by leaders who foster a culture able to change and invent the new".

Despite the contribution of innovation to society in a world of constant discovery, it is necessary that education and lifelong training, particularly higher education, keeps up with technological development, to qualify and instruct people with the necessary knowledge, so that they understand science and can make their professional, personal, and political choices [23]. According to Berikol B.Z., Killi M. [10] "ICT competencies are one of the basic technical skills required by accounting graduates. To meet this requirement and help students to be ready for working life, the accounting curricular programmes in Higher Education are required to include ICT software tools in accounting courses". Al-Htaybat et al. [24] presented a study focused on these expected changes and examined in an evaluation approach how the accounting profession, practice and, consequently, education will be affected by and adjust to these new technologies. They concluded significant changes are expected. Changes include amending respective courses to emphasise classic skills, such as problem solving, and contemporary skills, such as new technologies, to illustrate developments in a practical fashion.

National and international professional associations also play a crucial role in defining the profiles of skills that are needed to prepare professionals in the current context. Now, they are defining guidelines to update course curricula. Examples of this are the American Institute of CPAs (AICPA) and National Association of State Boards of Accountancy (NASBA). In 2021, they presented the AICPA CPA and the NASBA CPA Evolution Model Curriculum [24]. According to them, the curriculum covers the content necessary for all future CPAs (the Core- on Accounting and data analytics, Auditing and AIS, and Tax), as well as the three separate Disciplines (Business Analysis and Reporting, Information

Systems and Control, Tax Compliance and Planning). After analysing the programme, it can be seen that the AICPA Curriculum increases the coverage of the CPA exam in the area of IT, namely, information system and data science.

According to the World Economic Forum [25], because of DT, around 75 million jobs are under threat, "while 133 million new roles may emerge that are more adapted to the new division of labour between humans, machines and algorithms". To support their prediction, the World Economic Forum [26] identified the top 10 skills required for the years 2018 and 2022, as shown in Table 1.

**Table 1.** Comparison of demand for skills, 2018 versus 2022.

| Today, 2018 | Trending, 2022 | Declining, 2022 |
|---|---|---|
| Analytical thinking and innovation | Analytical thinking and innovation | Manual dexterity, endurance and precision |
| Complex problem-solving | Active learning and learning strategies | Memory, verbal, auditory and spatial abilities |
| Critical thinking and analysis | Creativity, originality and initiative | Management of financial, material resources |
| Active learning and learning strategies | Technology design and programming | Technology installation and maintenance |
| Creativity, originality and initiative | Critical thinking and analysis | Reading, writing, math and active listening |
| Attention to detail, trustworthiness | Complex problem-solving | Management of personnel |
| Emotional intelligence | Leadership and social influence | Quality control and safety awareness |
| Reasoning, problem-solving and ideation | Emotional intelligence | Coordination and time management |
| Leadership and social influence | Reasoning, problem-solving and ideation | Visual, auditory and speech abilities |
| Coordination and time management | Systems analysis and evaluation | Technology use, monitoring and control |

Source: (p.12, [27]).

Severini et al. [28] used the social accounting matrix in a multisectoral analysis to obtain a breakdown of work by level, education, digital skills, and gender, in Italy. The authors' findings suggest that countries with an abundance of highly skilled labour and digital skills are more developed. As a result, these countries tend to grow faster than others. Skills such as the ability to analyse data using business intelligence, data analysis, and exploration techniques, and synthesising and interpreting multiple sources, are some of the key skills required by employers. Not surprisingly, these are also some of the skills that the IMA Management Accounting Competency Framework [29] included in its accounting and management skills framework. Aiming to understand how current and future technologies are impacting accounting, Kruskopf et al. (p. 84–85, [30]) developed research where they identified what "kind of jobs might be offered in the future" and the "kind of employee companies will be looking for". Regarding the future jobs in the accounting field, Krauskopf et al. [30] identified eleven job titles: Blockchain Accountant, Analytics Guru, Historical Accounting Analyst, Healthcare Accountant, Cloud Accounting Specialist, Systems Integrator, Cybercrime Accountant, Fintech City Planner Accountant, Strategic Accounting Analyst, Fintech Accountant, and Data Security Accountant. Concerning the accounting profession, the authors added new skills to the base knowledge of accounting and broke the skills down into two categories, as shown in Table 2.

Many future accounting tasks will become a hybrid, i.e., they will be performed by professionals who interact with machines. In that sense, the mix of competencies will also be extended [4]. They will go far from the typical "preparation of income tax returns" of Chaplin (p. 61, [31]), the core accounting and management knowledge, and the already necessary soft skills of "analysis, innovative problem solving, communication and client relations" (p. 69, [32]), and they will include skills such as "data analytics skills and technology" (p. 33, [33]), "IT forensics, IT audit, and data analytics", (p. 166, [34]), and

"knowledge in transformation of new disclosure regulations, new forms of disclosures, and awareness of the interconnectedness of financial and non-financial reporting" (p. 2, [35]).

**Table 2.** Potential current and future skills required.

| Technical Skills/Hard Skills | | Social Skills | |
|---|---|---|---|
| Understanding the capabilities of the software | Basics of coding | Strong communication | Emotional intelligence ethical |
| Analysis skills | Fintech software knowledge | Conflict solving | Adaptability, tolerant of uncertainty |
| Data visualisation | Data security, forensic tools | Leadership skills | Sales knowledge |
| Knowledge of International Standard | Data warehouse management | Risk management | Innovative/Creative |
| Knowledge of industry-specific regulations | Enterprise resource planning (ERP) experience | Strategic decision making | Customer service orientation |

Source: (p. 85, [30]).

Finally, it is also important to recognise the influence of external pressures on the DT of organisations [36]. By introducing an institutional perspective to digital transformation [37], the literature reveals that many organisations initiate the digital transformation process as a reactive process to external pressures, without having a clear plan to approach it, thus compromising the results of their DT process. Indeed, all these internal and external dimensions must be analysed because, as stated by Fischer et al. (p. 2, [38]) "while companies can manage aspects of digitisation and digitalisation primarily by using information systems, they need to rely on holistic approaches when coping with the digital transformation".

## 3. Materials and Methods

To accomplish the study objective, a descriptive exploratory study with a qualitative approach was conducted. The methodology used was multiple case studies. Two cases concern software development companies, i.e., these are companies that are positioned in the middle of the value chain of the accounting business; they are back-office businesses because they do not deal with the final customer but sell their products and services mainly to accounting professionals and accounting companies. The third case concerns an accounting services and tax consulting company, i.e., a company that is positioned at the end of the business value chain, i.e., it is a front-office business because it sells its accounting services to individuals and companies.

### 3.1. Research Questions

Once the commitment to a qualitative research approach was established, the research question was framed in general terms as follows: How will digital transformation impact the future of the accounting sector?

To provide an operational answer to this question, a group of sub-questions should first be answered.

R1: How has Industry 4.0 been implemented in Portuguese accounting services SMEs?
R2: What are the main challenges and barriers to the implementation of Industry 4.0 in the accounting sector?
R3: How does Industry 4.0 impact accounting tasks/activities and the future of the accounting profession?
R4: How does Industry 4.0 impact the cybersecurity of accounting information?
R5: What are the digital skills of accounting professionals operating in the digital era?

### 3.2. Data Collection Process

Internal documents, Financial Reports from 2020, semi-structured interviews, and social media information were the main data sources. All our interviews and document collection practices meticulously observed the procedures suggested for exploratory qualitative research, particularly those specified for case studies [39] and interview processes [40].

The three companies under study were selected based on the following criteria: (1) companies should do business in IT solutions for accounting or tax and accounting services; (2) companies developed or used Industry 4.0 IT solutions for accounting; and (3) they are Portuguese.

The companies that are part of this study will be presented next, in addition to the technologies and solutions they offer to the market.

BiZDocs is an SME with 28 employees. It is a Revolutionary Document Archive Platform, simplifying the way companies around the world interact with all their business records in a General Data Protection Regulation (GDPR)-compliant environment. It provides a paperless accounting solution that transforms how businesses across the world interact with their accountants, and promotes such solutions as picture-perfect professional scanning of all business records in just one touch. The SME also provides easy access to all documents on any device, anywhere, powered by worldwide award-winning technologies (https://www.bizdocs.mobi, accessed on 12 May 2020).

Viseeon Portugal was established in 2020 following the merger of BTOC Consulting and Viseeon, an international financial consultancy network, and is characterized as a revolutionary model of accounting services, grounded in technology, to meet the challenges that accounting will bring. As the concept is based on network marketing, which allows growth as a freelancer, Viseeon provides accompaniment, independence, training, productivity, growth, and new resources and solutions, portraying harmony through the know-how of independent certified accountants and the contribution of new technologies. (https://www.viseeon.pt/, accessed on 14 May 2020).

IPBRICK by Expandindústria has, from the beginning, positioned itself as a full-service provider of company management support. The company presents technology-based solutions dedicated to the development of an ERP that aims at the analytical control of management in support of decision making, and is backed by the skills in technical assistance and training services. In May 2000, Expandindústria acquired IPBRICK with the main objective of enabling a more accessible and perceptible IT user experience. It currently has 41 employees (https://www.ipbrick.com/pt-pt, accessed on 18 May 2020).

The main IT solutions offered by each company are described in Table 3:

**Table 3.** The cases and IT solutions.

| Name | Partnership | IT | Products/Services |
|---|---|---|---|
| BizDocs | Latourrette Consulting | Robotic process automation, artificial intelligence, machine learning, optical character recognition | Digital archive of accountin documents. Cloud Sage for accountants. Digital archive. BizDocs accountants' portal |
| Viseeon Portugal | BTOC Consulting and International Viseeon | Robotic process automation, artificial intelligence, machine learning, optical character recognition | Financial and accounting consulting network and IT infrastructures sharing: Robot (Zé Manel); ERP; APP Intranet; APP Client. |
| IPBRICK | Expandindústria | Unified communications over IP on-premises, essentially in a private cloud, in collaboration with local data centres, to cover the most significant areas of business communications. | iPortalDoc—calls, emails and chat conversations are recorded in the system and are associated with the documents |

The CEOs of these leading companies in the industry, and three technical experts on their products/services, were interviewed to understand what has changed so far and how they see the profession developing. Interviews were conducted between July and August 2021. Table 4 depicts the professional data of the individuals included in the sample.

**Table 4.** Interviewees sample.

| Participant Code and Function | Training Area | Company | Business Sector | Professional Experience in the Area | Gender | Profile |
|---|---|---|---|---|---|---|
| P1 CEO | Accounting and Auditing | Viseeon Portugal, Lda. | Accounting Consulting | +20 years | M | Accounting Professional |
| P2 CEO | Degree in Marketing Master in Information Science | BizDocs by Latourrette Consulting | Software Company | +20 years | M | IT Manager |
| P3 CEO | Degree in Management Postgraduate in Auditing | Expandindústria-Studies, Projects and Management of Companies S.A. | Software Company | 20 | M | Accounting Professional |
| P4 Software Expert | Graduate in Electrical and Computer Engineering, Master in Industrial Automation PhD in Telecommunications | IPBRICK Expandindústria | Software Company | +20 | M | IT Manager |
| P5 Software Expert | Baccalauréat Scientifique Master Expert in informatics and information systems. | Viseeon Portugal, Lda. | Accounting Consulting | 6 | M | IT Manager |
| P6 Accounting Expert | Bachelor in Management and Information Technology | Visar Consultants Client of BizDocs | Accounting Consulting | 15 | F | Accounting Professional |

The semi-structured interviews were divided into three key stages, namely (1) the initial stage, where the objectives of the research study and ethical and legal considerations were mentioned, such as confidentiality and data disclosure; (2) the stage of the interview itself, consisting of previously established questions according to each interviewee's profile (accounting professionals and IT managers); and (3) the final stage, where the due acknowledgements and an informal conversation about the topic of the present study took place [41].

The interview scripts are summarised in Table 5.

**Table 5.** Summary of interview scripts.

| Section | Objective\Interviewee | |
|---|---|---|
| | **Company of Accounting Services** | **Company of Accounting Software Development** |
| Ethical and legal considerations | Presentation of the objectives of the interview and issues related to confidentiality and disclosure of data. | |
| | Identify the interviewee/company | |
| Auto-Perceptions and Personal Opinions | Familiarity with digital transformation and Industry 4.0. | The shift of paradigm in the accounting profession. |
| | Training and experience in technological innovation. | Diagnosis of the industry regarding digital transformation |
| Experience in DT processes | Motivations for the adoption of digital transformation. | Client's decision-making process regarding IT investments |
| | Digital Transformation process in accounting | Emerging IT used by the organisation. |
| | Advantages, challenges, and barriers to DT | Drivers for adopting new software |
| | Credibility and reliability of IT adopted solutions. | Tasks to be automated and other *software* functionalities. |

**Table 5.** *Cont.*

| Section | Objective\Interviewee | |
| --- | --- | --- |
| | **Company of Accounting Services** | **Company of Accounting Software Development** |
| Training and Competencies | Training the leader throughout his/her professional life. | Training needs and job obsolescence due to implementation of new IT |
| | Technological skills needed to handle software | Threat to accountants and other employees, fostered by the fear of losing their jobs to robotisation |
| Perceived Contributions | IT solutions adopted | Penetration in the market |
| | Gains in effectiveness, efficiency, error reduction | Contribution of accounting software for the competitiveness of the client, i.e., the accounting professional or company of accounting services |
| Expectations | Digital skills for accountants in the Digital Era. | Future of the accounting profession |
| | Expectations of DT in the accounting sector. | Expectations of DT in the accounting sector. |

After the interviews were conducted, the process of data transcription, analysis, and interpretation began. In the first stage, data analysis was performed individually, which consisted of reading and interpreting the questions asked and, subsequently, a joint analysis was performed to check whether there was a convergent vision concerning digital transformation.

## 4. Results and Discussion

For the presentation of the results of the interviews, six research questions were defined, considering the categorisation expressed in Table 5, which was the basis for the analysis of the interviews' content.

R1: How has Industry 4.0 been implemented in Portuguese accounting services SMEs?

As expected, all the IT managers were familiar with the terms and have graduate and postgraduate training. As far as managers are concerned, it was found that they have training in such areas as auditing and business management, but they do not have technological training, citing "*I didn't take nor do I have a course in Industry 4.0.*", "*It was a gradual process, which has allowed us to follow the processes very closely without the need for specific training*". However, the demonstrated a sensibility and knowledge on the topic in question.

All of the accounting professionals said that they had been involved in digital transformation processes, particularly in the accounting area—"*international professional experiences made me open my mind to what was happening in the world, to technological trends*", "*digital transformation is transversal to our society, and companies are one of the main drivers of this transformation*". There is undoubtedly great enthusiasm about technological evolution.

Concerning the implementation of Industry 4.0 in Portuguese companies of accounting services, there is the general perception that the accounting sector is divided into two main segments: the large enterprises (mainly international auditing, accounting, and tax consulting services), and small and medium-sized enterprises (mainly national accounting and tax services). The first segment is identified as early IT adopters; the second segment is identified as followers. Although large enterprises initiated the digital transformation a long time ago, the second group was pushed up by external forces, such as public services. Moreover, the small and medium-sized enterprises do not have technical or financial autonomy to sustainably develop their own IT solutions. As one of the accounting professional participants said: "*Nowadays, talking about Digital Transformation in large enterprises, such as PWC, makes no sense because it is trivial for them, but in small offices, it is something that is unthinkable, because technology is not always available at affordable prices, but the hardest thing of all is what to choose*". Despite the "*giant increase in productivity*" that DT provides to accounting services companies, all the participants, accounting professionals and IT managers, shared

the opinion that most of the small and medium-sized enterprises are still at the beginning, because "(...) *access to IT tools like robotics, artificial intelligence, data analysis, automatic reporting is expensive. We are talking about thousands of euros. Most offices can't afford that*". Within this last group, which represents the large majority of the accounting sector, tax authorities were identified as the key player. As an accounting professional said: "*the Tax Authority, which for decades was one of the main obstacles to technological evolution, became, in 2006 with the creation of "Simplex governance programme", one of the main levers of this whole process," democratising "digital transformation in companies, promoting the dematerialisation of processes and documents, creating increasingly comprehensive and intuitive digital platforms*". This is not a surprise since, in Portugal, accounting services are highly connected to tax consulting.

The impact of the tax authorities in the accounting sector was highly recognised by all participants:

> " . . . *when I finished my degree in Accounting, I realised that Portugal didn't have Accountants, but tax collectors, because nobody looked at Accounting as a management tool (...) when it came to Accounting, professionals talked about accounting codes and tax jargon*"—Accounting professional.

> " . . . *today, with the constant legislative change, mainly in tax and labour matters, this is one of the professions with the greatest need for training and upskilling . . .* ".—IT manager.

> "*the profession, seen from the outside, is always in strain, being pressured by the tax agenda, lots of manual work, deadlines imposed by the government that spoil plans, generating difficulty in seeing beyond what is planned . . . thus, it is not only about the monetary value, but also about the quality of professional and personal life that the technological evolution will bring to the lives of accounting professionals*"—IT manager and accounting professional.

These statements are in line with the idea that DT in accounting is driven by external forces [35,36].

Thus, Industry 4.0 has been implemented in Portuguese accounting services companies at two speeds: that of large enterprises vs. that of small and medium-sized enterprises. Despite this dual velocity, the vast majority of the Portuguese accounting services companies went digital in a reactive way to pressure from public entities.

The IT manager participants pointed out some solutions for the DT process, namely, using web environment applications and web drives (Google drive, Dropbox, etc.) to link accountants and businesses in a very simple way. They also talked about other solutions, namely the Azur platform, User Spirit, and OCR: "*OCR is part of an area of Artificial Intelligence that has several components that we use. The first component of Artificial Intelligence is something called Computer Vision, which is a machine that interprets and recognises characters ( . . . )*". These participants also mentioned RPA: "*Another area is RPA, which is an area of task optimisation that consists of simulating human activities* via *a machine, which can include, for example, going to a website, logging in, and downloading all the invoices straight away*". Another IT Manager said: "*The main emerging technology that will greatly simplify the tasks of accountants is electronic invoicing since the invoice's data is sent to the recipient, the tax authority, and of course to the accounting office. With the invoice data already in a processable format, it can, of course, be automatically entered into the accounting software of the accounting office*". Another said: "*Our goal is to automate as much as possible through intra-service communications (APIs), such as automatically retrieving tax information, sending notifications. When this intra-service communication is not possible, we use Robotic Process Automation (RPA) to enable robots to do time-consuming actions usually performed by humans. Then we use solutions developed by partners for document management, accounting production or other specific areas*".

R2: What are the main benefits, challenges, and barriers to the implementation of Industry 4.0 in the accounting sector?

The testimonies of all participants, accounting professionals and IT managers, reinforce the idea that it is connected to the main developments of the accounting sector since the last quarter of the 20th century, specifically regarding time-consuming tasks, unvalued

activities, and unintended errors. In line with Jasim [1], all the participants suggested that it simplifies accounting procedures while enhancing their efficiency and effectiveness, resulting in better opportunities for value creation activities:

> " ... *efficiency, which brings competitiveness and security to the information delivered".*

> " ... *free time from routine executions that, with fatigue, result in errors and decreased efficiency, and it is possible to transfer the accountants' time to more intelligent work with greater added value for the client".*

> " ... *saving man-hours and turning their time into smart time and adding value to the client".*

> " ... *a reduction in errors, provided that the software is properly implemented and parameterised, and more time for more refined processing of information".*

> " ... *the information is accessible in a clearer, more up-to-date way and, above all, at any time or place".*

> " ... *the information processing time has decreased considerably, as well as the error associated with the processing of that very same information".*

Overcoming human errors was a recurrent issue in the statements of all participants, accounting professionals and IT managers, thus confirming the argument of Mosteanu and Faccia [15] that robots perform better and faster than humans.

The dematerialisation of the accounting archive and the dynamic communication with the clients are other benefits attributed to DT, said accounting professionals: "(...) *the possibility of eliminating paper and dynamically managing all our clients' documentation, thus saving time to perform other tasks inherent to our profession". Moreover, clients "(...) don't have to go through paperwork. The client no longer needs to call asking for a certain invoice, they access digital archive".* Another participant, an IT manager, reinforced this same idea: "*with the digital archive, we no longer have lots of file folders, we can access the companies' documents anywhere and at any time without the need to go to the office; working from home is much easier since it is no longer necessary to take all the folders home".* The reduction in operational costs was also mentioned as a positive factor: "*the number of "physical" visits to the client has decreased considerably. Virtual contacts are now accepted as normal, which means more time and fewer costs".*

The main barriers identified by both IT managers and accounting professionals were price and resistance to change. However, price was cited as a problem of scale, i.e., depending on the size of the company. When approaching accounting enterprises of a considerable size, it was pointed out that "(...) *the problem is not the price, because it would probably be cheaper to operate with fewer employees, than to invest in software. The main barrier to adopting a more modern solution is always the resistance to change".* Another participant, an IT manager, pointed out that the software is not the big problem, i.e., the technical issue is not the main barrier: "*The surprise is always huge when they realise how quickly employees can adapt to new applications. And it's not only the speed of adaptation, but also the gains in productivity".* This point is clearly in line with [21,22], reinforcing the idea that "strategy is before technology" (p. 398, [22]). One of the other participants, also an IT manager, called attention to the problem of interconnection and interoperability of IT infrastructures [16].

Thus, digital transformation provides several benefits to Portuguese accounting services enterprises in terms of productivity, efficacy, and efficiency, releasing time from routine tasks to valued-added activities, reducing errors, and improving the quality of communication with the client. However, to obtain these benefits, organisations must guarantee the integration and interoperability of the IT solutions, and the adaptation of the organisational culture. The price is an important barrier for small and medium enterprises, but does not come first.

R3: How does Industry 4.0 impact accounting tasks/activities and the future of the accounting profession?

The automation of routine accounting tasks is a continuous process, and will have a great impact on the future of the day-to-day accounting profession. However, it is highly connected with the digital transformation of public entities and e-government strategies. Currently, there are many routine tasks performed by accountants that concern bureaucratic issues related to public entities, particularly the tax authority. As stated by one participant, an accounting professional, "*the main emerging technology that will greatly simplify the tasks of accountants is electronic invoicing since the invoice data is sent to Tax Authority ... no doubt the Periodic Declarations of VAT, IMI ... *". In this discourse, we noted the impact of DT in the future of the profession, in addition to the role of public entities, particularly the tax authority, as a key factor that externally pressures the organisation to institutionalise their IT solutions in a coercive way, which confirms the ideas proposed by Hinings et al. [37] and Yu and Pan [35].

From the analysis of the interviews, DT will impact the tasks and the professional profile as described in [2], [3], and [4]. One of the participants clarified this point by stating that "*the adoption of the software impacts at two levels: —the cultural change, i.e., to stop understanding accounting as just a system of registering documents between debits and credits for the preparation of information for third parties, namely banks and the tax authorities; shifting the relationship basis with clients to provide them with useful information for day-to-day decision-making. As information is only useful if it is produced and accessed in good time, DT introduces a new paradigm resulting from a new approach to accounting phenomena, namely the creation of a solid and consistent basis of multifaceted information which, among other objectives, may simplify and democratise access to information*". This means that DT will create the conditions for a much more analytic, creative, and value-added accounting profession in a context of abundant information, but also result in highly complex decision-making processes due to the increased difficulty of selecting the relevant information [7].

During the interviews, it became obvious that the accounting profession, as we know it today, will change. Professionals were described as follows: "*they will finally be accountants, because today they are just mappers, today they make maps (...). Today we fill in Tax Returns, maps to send to the client—some, others only send the balance sheet as it comes out of the system—and even then, only if the bank asks us to do so ... I believe that the manager and the accountant will work together on a closer basis, but they won't steal each other's work ... accountants perform their duties even better, and pay even more attention to their clients*". Therefore, as one accounting professional participant noted: "*Accounting enterprises that will manage to stay in business in the future will be those that offer added-value services to their clients, especially in Financial Management Consulting*". Again, the results of the interviews agree with the literature review. With technological developments and the handling of new technologies, partner managers can use them as a form of innovation, but also as a way of encouraging their staff to create new ways of using the working hours they gain to improve the service they provide to clients [14].

Indeed, as pointed out by one accounting professional participant: "*... now, it's not just technology. If you ask me: are people happy with robotics? They are, they are! Are they happy with automatic reporting? They are! ... but the satisfaction is much greater than that. Above all, they are happy. I give them clients because I save them a lot of time. Technology is a small part of it ... they are aware that they are providing a differentiated service, with more technology, more innovation and they are seeing something that many of them were not expecting: the value for clients is perceived. Many of them weren't expecting the positive feedback*". This fact was previously noted by Schwertner [22] in the literature review, reinforcing that technologies may considerably affect the market in the coming years. Forces such as cloud computing and data analytics, which are already revolutionary in themselves, when combined, will transform society as we know it, destroying obsolete business models and creating new leaders. Thus, organisations must keep abreast of technological developments so that they can evolve with the market, gaining not only market share, but also differentiation.

Thus, the expected changes are mainly in the routine tasks and tasks that accountants undertake due to the over-bureaucratic burden of public administration. Considering that

DT in accounting companies is driven by DT in public entities, it is expected that DT in public entities will determine the rhythm of DT in accounting. However, in essence, the role of the accounting profession in the organisation will mainly be the same, i.e., preparing useful information for decision making. However, the distributions of the working time will be highly impacted by DT, with much more time for creative, analytical, and value-added activities.

R4: How does Industry 4.0 impact the cybersecurity of accounting information?

The participants are aware of the cybersecurity issue regarding accounting information, and divided it into three broad categories, as indicated in the literature (p. 809, [19]). First, cybersecurity protects the confidentiality of private information [17]. This point of view was expressed by one IT manager participant in the following terms: "*the cybersecurity and data protection problems exist whether to more sophisticated or more old-fashioned applications. However, it is quite likely that older applications have more vulnerabilities than more modern ones ... of course, the fact that it is an innovative technology with online applications there are higher risks, but I believe they are credible and safe*". Second, cybersecurity ensures that authorised users can access information on a timely fashion [17]. This idea was well emphasised by all participants, for example: " *... access permissions are a crucial issue and this type of applications, when provisioned, also already foresee the safeguarding of the data against all kind of incidents ... *"; " *... we provide the network with cybersecurity training, and we have a series of safeguards and access controls to ensure that all access is appropriate ... *".

Third, cybersecurity protects the accuracy, reliability, and validity of information [17]. This last category was very well explained by one of the participants, an IT manager: " *... for software reliability, we include network members during the development phases so that we always have their input on developments and requirements. This ensures that we always have the expected functionality that respects data protection rules*".

In the literature review, it was also mentioned that the integrity and quality of the reported information is a challenge that comes with technological evolution [15]. However, according to an interviewee, "*(...) as long as there is this awareness, these tools are super reliable and are super compliant and I would say that they are even much more reliable and much more secure than those open folders that are often left at lunchtime on the accountant's desk or even overnight (...)*".

Thus, cybersecurity and data protection are cross-cutting problems, and accountants are aware of this. Regarding confidentiality, credibility, and reliability associated with the software, the dominant narrative expresses a high level of trust in the system, especially when compared to the existing security provided by the current information systems.

R5: What are the digital skills of accounting professionals operating in the digital era?

Even before DT occurred in the accounting sector, the accounting profession already covered several areas of knowledge, namely accounting, taxation, reporting, financial management, internal control, and information systems, in addition to problem solving, critical thinking, and communication competencies. All the participants recognised the importance of these core basic skills, in line with the skills described by Chaplin [31] and Howieson [32].

Although that knowledge basis will continue to be the same, the question that remains concerns the knowledge needed for the new task that will replace the old ones of collecting, checking, and preparing accounting data, i.e., that one task that will be automated. According to participants, accounting will be a call for more value-added activities, such as data analysis and management consulting, which will thus require the use of sophisticated IT tools. Consequently, according to the participants, IT managers and accounting professionals, future accounting professionals will be required to have specialised training in IT, as stated in the literature [32–34]. This means that, since "accountants are traditionally the reliable source for business information" (p. 33, [33]), they can take away the threat that hangs over their position due to the emerging profession of data analysts if they go digital without any resistance. The unresolved question was formulated by one participant, an accounting professional: "How am I going to adopt my team, my process, my structure?

Should I ask for help from the company that sells the software?" The answer was also given by the participant in the following terms:

*" ... software companies don't neglect training solutions ... We had to create mandatory online training on the tools we provide so that everyone can use them".*

*" ... they ought to go through to a training plan ... it's all free, it's all our investment ... then, the CEO and employees have 20 h of training per year free of charge, and they can buy more courses if they want, we have a wide informative offer. Of these 20 free hours, some courses are compulsory and, as they are free, we feel free to make attendance compulsory (...). Therefore, I would say that training is a false problem as it is compulsory, and you don't pay for it at all".*

*" ... the idea to start with digital archiving in the office came from the leadership (...), but I believe that all colleagues feel enthusiastic about this new future that is digital archiving ... this is another important field of training".*

Hence, according to the interviewee, in addition to the technical skills in the field of accounting, it is increasingly important that accountants can develop digital skills to take advantage of IT [4] and preserve their jobs [33].

The following quote helps us understand the mix of competencies and skills required for the future accounting professional: "*Accounting is taking its place as a science, a science that has measurement, processing, and communication as its pillars. The phenomenon to be measured, processed, and communicated (in the most diverse forms) are increasingly complex. Analytical or Management Accounting systems are fundamental for organisations to make decisions. Evaluating the performance of a particular department or section often involves communicating with a particular machine or equipment, meeting with teams of engineers and managers/accountants to redefine and adapt processes, define new metrics and evaluation systems. The accountant is in the middle of all these processes.*" Here, we find all of the technical and soft skills identified in Kruskopf et al. [30]. The importance of balancing these new skills with the traditional ones, without being over-enthusiastic about digital, was clearly stressed by one participant: "*... of course, if an accountant has digital skills and knows how to work with basic tools such as Excel, great. But what he has to know is how to look at the data, to have technical accounting skills that allow him to make correct data analysis, a correct sensitivity analysis, a correct projection of the same with the client and then he has to have many soft skills. You have to know how to communicate with the client, be empathetic, know how to involve the client so that they can think together, be able to study a little "accountancy", have more intelligent conversations and take advantage of the information (...). Now, unfortunately, an indirect answer to the question, what companies are asking for is just more and more technological skills. They are not putting the tools at their service, they are putting their technicians at the service of the tools and this, in my opinion, is very wrong*".

Thus, the digital skills of accounting professionals operating in the digital era are expected to combine the core tax, accounting, and management knowledge, and the already necessary soft skills of analysis, problem solving, communication, and client relations. The training in IT is expected to be a continuous process resulting from the partnership with an accounting software company.

Table 6 shows the main topics noted by the interviewees according to their profile (accounting professionals and IT managers).

**Table 6.** Topics noted by the interviewees.

| Sections | IT Managers | Accounting Professionals |
|---|---|---|
| Auto-Perceptions and Personal Opinions | • Experiences in the area of digital transformation; <br> • International experiences. | • Training in areas such as management and auditing. |

**Table 6.** *Cont.*

| Sections | IT Managers | Accounting Professionals |
|---|---|---|
| Experience in DT processes | Motivations and advantages:<br>• Productivity;<br>• Greater autonomy to reduce routine tasks;<br>• Decreased workload;<br>• Quality of results;<br>• Reduction in errors;<br>• Safe and reliable tools;<br>• The safeguard of data against incidents;<br>• Respect for data protection rules.<br>Challenges:<br>• Price;<br>• Resistance to change. | Motivations and advantages:<br>• Efficiency;<br>• Competitiveness;<br>• Decreased workload;<br>• Rigour and security of the information;<br>• Sense of proximity;<br>• Dynamic management of customer documentation;<br>• Automation of routines and tasks;<br>• Accessible information in a clearer and more up-to-date manner;<br>• Decrease in the number of physical displacements and paper disposal;<br>Challenges:<br>• Starting the process;<br>• Lack of willingness to change to new;<br>• Price. |
| Training and Competencies | • Mandatory and free training solutions. | • Quick adaptation;<br>• Easy handling;<br>• Enthusiasm of colleagues about digital archiving. |
| Perceived Contributions | • The profession is always under pressure;<br>• Automation of routine accounting tasks;<br>• Positive perceived contribution. | • Positive degree of satisfaction;<br>• More customers and administrative time saved;<br>• Free time for data analysis. |
| Expectations | Skills<br>• A simpler learning curve when compared to previous applications;<br>• All the hard work will disappear;<br>• Evolution in the type of tasks or services offered;<br>• Coexistence between managers and accountants;<br>• Loss of jobs. | Skills<br>• Society responsible for giving digital skills to candidates;<br>• More and more technical skills are in demand;<br>• Placing technicians at the service of the tools;<br>• The profession is already changing;<br>• Technology is needed to free them from repetitive tasks;<br>• Coexistence between managers and accountants;<br>• Loss of jobs. |

## 5. Conclusions

The DT of accounting is an ongoing process with a great impact on the accounting information systems of organisations, on accounting as an economic sector, and on the accounting profession itself. This study aimed to analyse the impact of digital transformation in the accounting sector, with special attention paid to accounting professionals. An exploratory approach and multiple case studies were adopted to gather broader empirical evidence.

The study shows that, although digital transformation is at the beginning stages in Portuguese accounting services SMEs, in terms of emerging technologies, OCR, AI, and the cloud are at the top of the list of the adopted IT. The resistance to change, the organisational culture, and the price seem to be the main barriers to DT in accounting. In terms of

advantages, the automation of routine tasks and the reduction in errors were unanimously recognised. These advantages release time for accountants to perform higher value-added services and eliminate paper. In this sense, in the field of accounting, the man–machine interaction may be harmonious and human adaptability as a social competence may bridge the desired gap. The cybersecurity and data protection related to accounting data are cross-cutting problems, and accountants are aware of this. Digital skills of accounting professionals operating in the digital era are expected to be combined with the already required knowledge, competencies, and skills, in a logic of adding and substituting.

This paper is exploratory; thus, its main contribution is to clarify the discussion about the impact of digital transformation in the accounting sector, and it does not prove any causal effect. Nonetheless, the results can be of great interest to researchers, policymakers, teachers, professional bodies, and accounting professionals.

However, the study has important limitations. The impact of DT on the accounting sector is extremely complex to assess. It is mediated by a myriad of uncontrollable variables. In addition to the limitation intrinsic to qualitative research, the main limitation of the study is the dimension of the sample.

For further research purposes, it is proposed that the study should be replicated in other national and international companies. It is also proposed that a quantitative study of the implementation of Industry 4.0 in accounting sector companies should be carried out.

Once this research work is concluded, it is expected that it may contribute to future works and knowledge enrichment, at the level of digital transformation and the development of business.

**Author Contributions:** Conceptualization, M.J.A.G., A.C.F.d.S. and C.G.F.; Data curation, C.G.F.; Funding acquisition, M.J.A.G.; Investigation, C.G.F.; Project administration, M.J.A.G.; Supervision, M.J.A.G. and A.C.F.d.S.; Writing—original draft, C.G.F.; Writing—review & editing, M.J.A.G., A.C.F.d.S. and C.G.F. All authors have read and agreed to the published version of the manuscript.

**Funding:** This work is financed by Portuguese national funds through FCT—Fundação para a Ciência e Tecnologia, under the project UIDB/05422/2020.

**Institutional Review Board Statement:** Not applicable.

**Informed Consent Statement:** Informed consent was obtained from all subjects involved in the study.

**Data Availability Statement:** https://zenodo.org/record/6298850#.YhoNY6vP25c, accessed on 18 May 2020.

**Conflicts of Interest:** The authors declare to have no conflict of interest. The funders had no role in the design of the study; in the collection, analysis, or interpretation of data, in the writing of the manuscript, or in the decision to publish the results.

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
