# Peer review of "The Future of Accounting: How Will Digital Transformation Impact the Sector?"

_informatics, doi:10.3390/informatics9010019_

Round 1

Reviewer 1 Report

Using a qualitative and explorative approach through cases studies of two accounting software developers and one accounting firm in Portugal, this paper attempts to understand how the financial and accounting services industry perceives the impact of the digital transformation on the accounting profession. Given that the information technology is expected to drastically change the landscape of the accounting service industry, it is of great interest to gain insight into how accounting professionals respond to such sea changes.

While the paper is overall well-done and provides interesting perspectives from the accounting practitioners regarding this pertinent issue, I strongly suggest the paper be thoroughly proof read for proper English including word choices and sentence structure. My other comments follow:

  • A little more background of the companies in the case study would be appreciated. These companies may be leaders in their industry in Portugal, but they maybe unfamiliar to those of us non-Portugese. Some background about their industry position including their size can help a reader gauge the impact of their opinion and which segment of the industry the represent.
  • The same comment goes for the background of the interviewees. The paper presents their educational background, which is important to the research question. However, as a case study, I would also be interested in knowing more about the interviewees including years of experience in the industry, gender, previous positions held, etc.
  • The Appendix lays out a nice comparison and contrast between the views of the accounting software developers and the accounting firm. The authors could have done the same in their narratives to highlight the similarities and difference between those two groups. It’s not always clear to me which group the paper is quoting, as I assume they would have somewhat different perspectives.

Author Response

Thank you very much for your consideration.

Reviewer 1

Reviewer Comments

Revision

While the paper is overall well-done and provides interesting perspectives from the accounting practitioners regarding this pertinent issue, I strongly suggest the paper be thoroughly proof read for proper English including word choices and sentence structure.

The language was improved.

A little more background of the companies in the case study would be appreciated. These companies may be leaders in their industry in Portugal, but they maybe unfamiliar to those of us non-Portugese. Some background about their industry position including their size can help a reader gauge the impact of their opinion and which segment of the industry the represent.

The companies were characterized

The same comment goes for the background of the interviewees. The paper presents their educational background, which is important to the research question. However, as a case study, I would also be interested in knowing more about the interviewees including years of experience in the industry, gender, previous positions held, etc.

The interviewees were described. The data of the study will be available on an open-source platform.

The Appendix lays out a nice comparison and contrast between the views of the accounting software developers and the accounting firm. The authors could have done the same in their narratives to highlight the similarities and difference between those two groups. It’s not always clear to me which group the paper is quoting, as I assume they would have somewhat different perspectives.

The results and analysis of the study have been restated.

The table was changed and it is in the discussion section.

Reviewer 2 Report

This is a good paper. It does a good job of covering cloud computing, artificial intelligence, dig data, and data analytics as emerging technologies in accounting. 

To improve this paper, I suggest that the authors include information about the new AICPA Curriculum for the revised CPA exam starting in 2024.  Perhaps note how the AICPA Curriculum Map increases the coverage of the CPA exam in the areas of information system controls over financial reporting. Also, consider noting that two of the three new Disciplines for the new CPA Exam are "Information Systems Controls" and "Data Analytics." 

The authors might also consider adding a section on System and Organization Controls (SOC) for third-party engagements performed under Statement on Standards for Attestation Engagements 18 (SSAE 18) and how CPAs require knowledge of information system controls as well as controls over security, availability, processing integrity, confidentiality, and privacy controls described in the AICPA’s Trust Services Criteria.

Author Response

Reviewer 2

Thank you very much for your comments.

Reviewer Comments

Revision

To improve this paper, I suggest that the authors include information about the new AICPA Curriculum for the revised CPA exam starting in 2024.  Perhaps note how the AICPA Curriculum Map increases the coverage of the CPA exam in the areas of information system controls over financial reporting. Also, consider noting that two of the three new Disciplines for the new CPA Exam are "Information Systems Controls" and "Data Analytics." 

The state of the art was updated and recommendations for updating the curriculum were introduced, namely the revised AICPA Curriculum for the CPA examination.

The authors might also consider adding a section on System and Organization Controls (SOC) for third-party engagements performed under Statement on Standards for Attestation Engagements 18 (SSAE 18) and how CPAs require knowledge of information system controls as well as controls over security, availability, processing integrity, confidentiality, and privacy controls described in the AICPA’s Trust Services Criteria.

The audit perspective, in our opinion, is also very important, however; the focus of the article was to analyse the Accounting processes.

The language was improved.

Reviewer 3 Report

This article aims to show the influence of the digital transformation on accounting. Five companies were examined and a total of six interviews were conducted. Overall, the topic can be classified as very relevant, but the article remains somewhat superficial and provides only few new insights. 
The literature review could be expanded and backed up with more sources. Some of the analysed companies seem to use very advanced technologies (RPA, AI, machine learning) etc.. However, there are hardly any results from the interviews on how these technologies actually perform in practice.  What are the consequences of pointing out that these tools are very expensive for small- to medium-sized companies? All in all the number of companies interviewed could be expanded. It would also be interesting to know what long-term vision the people interviewed have for the accounting sector. The required digital competences could be described more in detail and maybe also depending on the different roles in accounting (e.g. financial accountant, tax expert, management accountant, business data analyst, ...). 

Author Response

Thank you very much for your comments.

Reviewer Comments

Revision

This article aims to show the influence of the digital transformation on accounting. Five companies were examined and a total of six interviews were conducted. Overall, the topic can be classified as very relevant, but the article remains somewhat superficial and provides only few new insights.

The literature review could be expanded and backed up with more sources. Some of the analysed companies seem to use very advanced technologies (RPA, AI, machine learning) etc.. However, there are hardly any results from the interviews on how these technologies actually perform in practice.  What are the consequences of pointing out that these tools are very expensive for small- to medium-sized companies? All in all the number of companies interviewed could be expanded. It would also be interesting to know what long-term vision the people interviewed have for the accounting sector. The required digital competences could be described more in detail and maybe also depending on the different roles in accounting (e.g. financial accountant, tax expert, management accountant, business data analyst, ...).

The bibliography has been updated.

The presentation and discussion of the results section has been updated

Regarding the number of interviews, according to Creswell and Poth (2016), a minimum of 5 interviews and a maximum of 30 interviews is adequate for qualitative research. However, we agree with the reviewer that "the number of companies interviewed could be expanded".  We will, over a period of 2 years, carry out a longitudinal study of the companies. We currently have post-graduate students studying accounting processes, monitoring the implementation of IT in these processes, and contributing to process improvement, using action research methodology.

Round 2

Reviewer 2 Report

Excellent modifications to the paper, including new sections on AICPA's new model curriculum.  I recommend accepting this paper for publication

Reviewer 3 Report

The paper was greatly improved and my comments were sufficiently taken into account. Thus, a publication can be recommended.